# Role of the *ADAM33* rs2280091 Variant in Modulating Lung Function in Cystic Fibrosis

**DOI:** 10.3390/ijms262311583

**Published:** 2025-11-29

**Authors:** Vinícius Santiago dos Santos, Lucas Silva Mello, Luiz Felipe Azevedo Marques, Luana Rodrigues Silva, Carmen Sílvia Bertuzzo, José Dirceu Ribeiro, Fernando Augusto Lima Marson

**Affiliations:** 1Laboratory of Molecular Biology and Genetics, Postgraduate Program of Health Sciences, Postgraduate Program of Health Data Science, University of São Francisco (Universidade São Francisco—USF), Bragança Paulista 12916-900, SP, Brazil; vinicius.santiago.santos@mail.usf.edu.br (V.S.d.S.); lucas.silva.mello@mail.usf.edu.br (L.S.M.); luiz.azevedo@mail.usf.edu.br (L.F.A.M.); luana.ssouza@mail.usf.edu.br (L.R.S.); 2Laboratory of Clinical and Molecular Microbiology, Postgraduate Program of Health Sciences, Postgraduate Program of Health Data Science, University of São Francisco (Universidade São Francisco—USF), Bragança Paulista 12916-900, SP, Brazil; 3LunGuardian Research Group—Epidemiology of Respiratory and Infectious Diseases, Postgraduate Program of Health Sciences, Postgraduate Program of Health Data Science, University of São Francisco (Universidade São Francisco—USF), Bragança Paulista 12916-900, SP, Brazil; 4Department of Pediatrics, University of Campinas (Universidade de Campinas—Unicamp), Campinas 13083-970, SP, Brazil; bertuzzo@unicamp.br (C.S.B.); jdirceuribeiro@gmail.com (J.D.R.)

**Keywords:** airway remodeling, cystic fibrosis phenotype, genetic association studies, modifier genes, polymorphism, genetic, pulmonary function tests

## Abstract

Cystic fibrosis (CF) is a rare genetic disease caused by pathogenic variants in the *CFTR* (*Cystic Fibrosis Transmembrane Conductance Regulator*) gene, with wide clinical variability influence not only by the *CFTR* genotype but also by environmental and modifier genes such as *ADAM33* (*A Disintegrin and Metalloproteinase Domain 33*). The rs2280091 variant in *ADAM33* may affect lung function and contribute to differences in disease severity. This study investigated the association between this genetic variant and lung function in CF patients. This cross-sectional study included 55 CF patients from a Brazilian center, with diagnosis confirmed by sweat testing and *CFTR* genotyping. Pulmonary function was evaluated by spirometry before and after bronchodilator (BD) administration according to the American Thoracic Society/European Respiratory Society guidelines, analyzing Forced Vital Capacity (FVC), Forced Expiratory Volume in one second (FEV1), FEV1/FVC ratio, Forced Expiratory Flow at 25%, 50%, and 75% of FVC (FEF25%, FEF50%, and FEF75%), mean Forced Expiratory Flow between 25% and 75% of FVC (FEF25–75%), and Maximal Expiratory Forced Flow (MEF). The *ADAM33* rs2280091 variant was genotyped by polymerase chain reaction-restriction fragment length polymorphism (PCR-RFLP), and statistical analyses included Kruskal–Wallis and Mann–Whitney tests, chi-square (χ^2^) tests, and calculation of odds ratios (ORs) with 95% confidence intervals (95% CI). The study included 55 CF patients, predominantly female (96.4%) and Caucasian (52.7%), with a median age of 17 years. *CFTR* genotyping revealed F508del/F508del as the most common genotype (52.7%). Analysis of the *ADAM33* rs2280091 variant demonstrated that the AA genotype was most frequent in both CF patients (69.1%) and healthy controls (78.6%). Notably, the GG genotype was significantly enriched in CF patients (18.2%) compared with the controls (0.02%), yielding an odds ratio of 12.06 (95% CI: 4.86–29.91), while the G allele was also associated with increased disease risk (24.5% vs. 11.6%). Pulmonary function assessment indicated that carriers of the GG genotype or G allele had higher Forced Expiratory Flow parameters (FEF25%, FEF50%, FEF25–75%, and MEF) and improved BD responsiveness, suggesting a potential modulatory role of *ADAM33* in peripheral airway function in CF. The G allele of the *ADAM33* rs2280091 variant was more frequent among recruited CF patients and associated with improved peripheral airway function and BD response. These findings may reflect a survivor effect, in which carriers of this allele are more likely to reach clinical follow-up and recruitment rather than indicating a direct association with increased disease risk.

## 1. Introduction

Cystic fibrosis [CF, Online Mendelian Inheritance in Man (OMIM) no. 219700] is a rare autosomal recessive genetic disease characterized by exocrine gland dysfunction and multisystem involvement, primarily affecting the respiratory and digestive tracts [1,2,3]. It results from pathogenic variants in the *CFTR* (Cystic Fibrosis Transmembrane Conductance Regulator) gene [4,5,6,7]. The *CFTR* gene encodes an Adenosine Triphosphate (ATP)-dependent ion channel protein that is essential for chloride and bicarbonate transport across epithelial surfaces. Alterations in this protein impair the ionic and water homeostasis of secretions, leading to the production of thick and viscous mucus [6,8,9,10]. This dysfunction promotes, for instance, airway obstruction, persistent bacterial colonization, and chronic inflammation—key factors in the pathophysiology and progression of CF [4,5,6,11,12].

The average incidence of CF in Brazil is estimated at 1 in 10,000 live births, being more prevalent among individuals of European ancestry [13]. In Latin America, recent studies report prevalence rates ranging from 1 in 8000 to 1 in 10,000, reflecting the region’s genetic diversity [14]. With the advancement of newborn screening programs, many cases have been diagnosed earlier, enabling prompt clinical interventions [4,6,8,13,15]. Pulmonary involvement remains the leading cause of morbidity and mortality in CF, characterized by progressive decline in respiratory function, recurrent infections, and acute exacerbations that often require hospitalization—highlighting the importance of individualized clinical management strategies [9,16]. In recent years, however, the introduction of CFTR modulators has revolutionized disease management, leading to significant improvements in pulmonary function, nutritional status, and overall survival [10,17]. Nevertheless, clinical variability among individuals carrying identical genotypes continues to represent a major challenge [18,19,20].

Despite therapeutic advances and the significant improvement in the prognosis of CF, pulmonary complications remain the leading cause of morbidity and mortality among patients [13,17,19,20,21]. In this context, comprehensive assessment of pulmonary function is essential for early diagnosis, clinical severity stratification, and the development of individualized therapeutic strategies [6,9,10,21].

Recent studies have demonstrated that the clinical heterogeneity observed in CF cannot be explained solely by *CFTR* genotype [22,23,24]. Growing evidence indicates that this variability arises, in part, from the interaction between environmental factors and the action of modifier genes, which can modulate the phenotypic expression of the disease [25,26]. These genes do not directly alter the primary *CFTR* mutation but influence the clinical course, affecting the severity of pulmonary involvement, the inflammatory response, susceptibility to respiratory infections, and even therapeutic responsiveness [20,27]. Understanding this complex genetic landscape is crucial to elucidate the mechanisms underlying disease progression and identify potential targets for personalized therapies [18,19,27,28,29,30,31,32,33]. Among the most extensively studied modifier genes is *ADAM33* (A Disintegrin and Metalloproteinase Domain 33), located on chromosome 20p13. This gene encodes a metalloproteinase expressed in fibroblasts and bronchial smooth muscle cells, involved in airway remodeling, cell proliferation, adhesion, and inflammation, playing a significant role in chronic respiratory diseases [34,35].

Variants of the *ADAM33* gene have been widely associated with bronchial hyperresponsiveness, persistent inflammation, and a progressive decline in lung function in respiratory diseases such as asthma [34,36]. Among these, the rs2280091 (NC_000020.11:3669586:A>G) variant has received particular attention due to its association with altered pulmonary function, increased airway inflammation, and airflow obstruction in individuals with asthma [35,36,37]. However, its role in CF remains under investigation. Considering that the pathophysiology of CF also involves chronic inflammation and pulmonary remodeling, it is relevant to explore the potential contribution of this variant to the clinical heterogeneity of the disease [34,38,39].

Understanding the influence of *ADAM33*, particularly the rs2280091 variant, on lung function in patients with CF may provide valuable insights into the genetic mechanisms underlying phenotypic modulation. This analysis opens new perspectives for patient stratification based on genetic profiles, enabling the development of more effective and targeted therapeutic approaches. Thus, the rs2280091 variant emerges as a potential target for personalized interventions in the management of CF.

## 2. Results

### 2.1. Clinical, Anthropometric, and Genetic Characterization of Participants

Table 1 provides a detailed characterization of the participants with CF included in the study, encompassing sociodemographic, anthropometric, and genotypic variables. The sample consisted predominantly of females (96.4%) and self-declared Caucasians (52.7%). The median age was 17 years (interquartile range: 14–21 years), representing a predominantly young population. Anthropometric parameters indicated short stature and low body weight, with median values of 48.70 kg for weight, 1.59 m for height, and 17.89 kg/m^2^ for body mass index (BMI), consistent with the clinical phenotype of CF. Regarding *CFTR* genotyping, the most frequent genotype was the homozygous F508del/F508del (52.7%), followed by compound heterozygotes such as F508del/G542* (16.4%) and other less common combinations. These findings highlight the genotypic heterogeneity of the sample and its relevance to the clinical profile of the disease.

### 2.2. Distribution of CFTR Gene Variants

Table 2 describes the *CFTR* gene variants identified in the study sample, including traditional nomenclature, allele count, relative frequency, complementary Deoxyribonucleic Acid (cDNA) and protein alterations, functional classification, Single Nucleotide Polymorphism database identifier (dbSNP identifier), and prevalence according to data from the Cystic Fibrosis Foundation (CFF) and the Brazilian Cystic Fibrosis Registry (Registro Brasileiro de Fibrose Cística—REBRAFC).

The F508del variant was the most prevalent, accounting for 56.30% of alleles. It is classified as a class II variant, associated with the absence of functional protein due to defective processing and trafficking. Other relevant variants included G542* (7.62%), N1303K (3.23%), and R1162* (2.64%), distributed across functional classes IA, IB, II, and IV, reflecting distinct pathogenic mechanisms such as impaired protein synthesis, premature truncation, and folding defects. Comparison with data from the CFF and REBRAFC registries provides epidemiological context for the variant frequencies observed in this cohort, reinforcing the importance of *CFTR* genotyping for both clinical management and therapeutic decision-making in CF.

### 2.3. Genotypic Frequency of the rs2280091 (ADAM33) Variant

Table 3 presents the genotypic distribution of the rs2280091 variant of the *ADAM33* gene among participants with CF and healthy controls, along with the statistical association analysis using the odds ratio (OR) and 95% confidence interval (95% CI).

The rs2280091 variant deviated from Hardy–Weinberg equilibrium in the CF group [χ^2^ test, Online Encyclopedia for Genetic Epidemiology Studies (OEGE) platform; *p* < 0.05], while the healthy control group remained in equilibrium (*p* > 0.05). This deviation was explicitly recorded in Table 3 and suggests either a true association with disease status, population stratification, or potential genotyping bias; possible explanations and implications are discussed further in Section 3.

The AA genotype was the most frequent in both groups (69.1% in cases and 78.62% in controls), showing no statistically significant association. In contrast, the GG genotype exhibited a markedly higher frequency in CF cases (18.2%) compared with controls (0.02%), yielding an OR of 12.06 (95% CI: 4.86–29.91), indicating a strong association with CF (*p*-value < 0.001). Genotypic grouping analyses (AA vs. AG + GG) and allelic frequency comparisons (A vs. G) further supported the association of the G allele with an increased risk of disease, demonstrating robust statistical significance (*p*-value < 0.001).

### 2.4. Association Between the rs2280091 (ADAM33) Variant and Pulmonary Functional Parameters

The combined analysis of data presented in Table 4 and Table 5 demonstrated a significant association between the genotypes and alleles of the *ADAM33* rs2280091 variant and spirometric parameters related to peripheral airway flow in patients with CF. Although most clinical and anthropometric markers did not show statistically significant associations with the genotypes (AA, AG, GG) or alleles (A, G), spirometric parameters—particularly those reflecting Forced Expiratory Flow—exhibited relevant differences among genetic groups.

#### 2.4.1. Genotype-Specific Findings

**GG genotype:** Individuals carrying the GG genotype exhibited the most favorable spirometric performance across the cohort, consistently outperforming the AA and AG groups in parameters reflecting small and medium airway function. Pre-bronchodilator predicted Forced Expiratory flow at 25% of forced vital capacity (FVC) (FEF25%) reached a median of 82% [interquartile range (IQR): 50–117.25], whereas AA and AG carriers demonstrated substantially lower values of 43% (IQR: 22.25–76) and 51% (IQR: 15–56), respectively. A similar trend was observed post-bronchodilation, with GG participants achieving median FEF25% values of 78% (IQR: 56.50–137.50), while AA and AG remained at 50% (IQR: 21–80) and 55.5% (IQR: 18.75–62.50).

For predicted forced expiratory flow at 50% of FVC (FEF50%), which represents airflow at mid-expiration and is sensitive to small-to-medium airway obstruction, GG carriers demonstrated median post-bronchodilator values of 48% (IQR: 41–121), exceeding those of AA (24%; IQR: 12–62) and AG (26.50%; IQR: 10.50–41.75). Similarly, Maximal Expiratory Forced flow (MEF) showed a marked gradient favoring GG carriers, with post- bronchodilator values of 84% (IQR: 66–124.50) compared with 61% (IQR: 45–85) in AA and 55% (IQR: 45–85) in AG. Collectively, these results suggest that individuals with the GG genotype experience less peripheral airway compromise and a more pronounced bronchodilator response, consistent with a functionally advantageous physiological profile.

**AG and AA genotypes**: Participants with AG and AA genotypes demonstrated consistently lower values for nearly all flow-dependent spirometric indices. Post- bronchodilator Forced Expiratory Flow between 25% and 75% of FVC (FEF25–75%), an index capturing mean flow throughout the mid-portion of the forced maneuver, reached 47% (IQR: 35–110) in GG carriers but only 27.5% (IQR: 12.75–64) and 17% (IQR: 6.5–24.25) in AA and AG, respectively. These differences were directionally consistent for all assessed parameters, reinforcing a genotype-dependent gradient in peripheral airflow performance. These findings suggest that the presence of the A allele may predispose patients to greater airflow limitation, while the G allele may attenuate peripheral airway narrowing.

#### 2.4.2. Allelic Comparison

Allelic analysis further supported the genotype-level observations. When comparing alleles directly, carriers of the G allele demonstrated significantly better lung function performance in parameters reflecting peripheral airway caliber and bronchodilator response. For example, post-bronchodilator FEF25% reached a median of 64% (IQR: 56–125.75) in G-allele carriers compared with 51% (IQR: 21–75) in A-allele carriers [*p*-value = 0.006, false discovery rate (FDR)-adjusted]. Similar differences were observed for FEF50% (47% vs. 24%; *p*-value = 0.015, FDR-adjusted), reinforcing the notion that the G allele is associated with superior functional performance.

Additionally, differences were apparent even before bronchodilation, such as pre-bronchodilator FEF25% (56% vs. 44%; *p*-value = 0.012, FDR-adjusted), indicating that the functional advantage conferred by the G allele may be present independently of acute bronchodilator response. Notably, the FEV1 (Forced Expiratory Volume in one second of FVC)/FVC ratio also differed significantly between alleles both pre- and post-bronchodilator (*p*-value = 0.036, adjusted), highlighting that the impact of *ADAM33* may extend beyond terminal airway function to mechanisms affecting airflow preservation across the respiratory cycle.

These findings indicate that the G allele of rs2280091 in *ADAM33* may be associated with better airflow dynamics and bronchodilator responsiveness in CF patients. However, rather than reflecting a direct disease association, this effect may relate to overlapping airway remodeling mechanisms also described in asthma and to a possible survivor effect, in which carriers of the G allele are more likely to reach clinical follow-up and study recruitment.

Appendix A illustrates the molecular and clinical complexity of CF and its genetic modulation. CF is a multisystemic disorder caused by mutations in the *CFTR* gene, leading to defective ion transport, thickened mucus, and chronic infection, particularly in the respiratory tract. The figure highlights the main systemic manifestations, including pulmonary disease, pancreatic insufficiency, liver dysfunction, intestinal obstruction, and male infertility. It also summarizes the seven functional classes of *CFTR* mutations, which differ in their impact on protein synthesis, processing, and channel activity, thus contributing to phenotypic heterogeneity. Finally, it depicts the modulatory effect of modifier genes, such as *ADAM33* (rs2280091), which may influence airway inflammation and tissue remodeling, partially explaining interindividual variability in disease severity among patients with similar *CFTR* genotypes.

Only spirometric parameters that remained statistically significant after multiple testing correction were included in the figures. Comparison of lung function across *ADAM33* rs2280091 genotypes showed differences in selected spirometric measurements (Figure 1 and Figure 2).

## 3. Discussion

This study evaluated the frequency and impact of the rs2280091 variant of the *ADAM33* gene on pulmonary function in 55 patients with CF treated at a Brazilian referral center and compared them with 608 individuals without CF from the Online Archive of Brazilian Mutations (ABraOM) database [40]. The study population consisted predominantly of young female participants of self-declared Caucasian ethnicity, with low body weight and height, and showed marked heterogeneity in *CFTR* genotypes, with F508del/F508del being the most frequent. With respect to *ADAM33* rs2280091 distribution, the AA genotype was the most prevalent in both groups; however, the GG genotype demonstrated the highest statistical relevance by occurring disproportionately more often in CF patients than in the controls. Importantly, the rs2280091 variant deviated from Hardy–Weinberg equilibrium in the CF group, which may reflect recruitment-related factors or a survivor effect, rather than a true disease-association signal. Carriers of the G allele showed higher spirometric values, especially in indices reflecting peripheral airway flow, indicating a possible influence of this allele on airflow dynamics and bronchodilator responsiveness. These findings support the potential role for *ADAM33* rs2280091 as a modifier gene contributing to phenotypic variability in CF while emphasizing the need to interpret allele frequency shifts within the context of Hardy–Weinberg equilibrium deviation.

An important factor contributing to the heterogeneity of pulmonary function in patients with CF is environmental exposure [43]. Although air pollution was not directly assessed in this study, previous research has demonstrated that external environmental factors can modify airway inflammation and clinical outcomes in CF. For example, in a cohort of 11,484 patients, Goss et al. (2004) reported that a 10 μg/m^3^ increase in particulate matter was associated with an 8% higher likelihood (95% CI: 2–15%) of experiencing more than two pulmonary exacerbations per year [44], underscoring that airway function in CF is influenced not only by genetic background but also by environmental burden. This supports the rationale for considering both intrinsic factors, such as *ADAM33* variation, and external exposures when interpreting pulmonary phenotypes in CF.

Also, among the possible reasons for the differences observed in pulmonary function and clinical severity among patients with CF, the genetic variability of *CFTR* mutations stands out [25,45,46,47]. In some populations, such as the Brazilian one, this variability is remarkably extensive [12,45,48,49]. Pereira and colleagues (2019) identified 63 variants—classified into seven distinct groups and 77 different genotypes—in a sample of 169 patients with CF [50]. Some of these variants were not listed in the databases of the CFF or the REBRAFC, highlighting the considerable genotypic heterogeneity of the disease and the potential underreporting of specific genotypes [50]. In addition to the analysis of pathogenic *CFTR* variants, recent studies have emphasized the role of modifier genes in determining the clinical phenotype of CF [22,23,24,26,38,51]. In this context, our findings support that the rs2280091 variant of *ADAM33* may act as one such modifier, helping explain differences in pulmonary function even among individuals with similar *CFTR* genotypes.

Among the modifier genes associated with CF, several have been investigated for their potential influence on disease severity [22,52,53,54,55,56,57,58,59]. These include members of the *SLC* family (*Solute Carrier family*), which encode membrane transport proteins responsible for ion and metabolite transport, thus influencing ionic homeostasis and acid–base balance [60,61,62]; *ACE2* (*Angiotensin-Converting Enzyme 2*), which regulates blood pressure, modulates inflammation, and provides epithelial protection in the lungs [63]; *GSH* (*Glutathione synthetase*), a key enzyme in glutathione synthesis, the major intracellular antioxidant [64,65]; *GCLC* (*Glutamate–Cysteine Ligase Catalytic Subunit*), the catalytic subunit of the rate-limiting enzyme in glutathione biosynthesis, essential for maintaining redox balance [64,65]; *ADRB2* (*Adrenoceptor Beta 2*), a β_2_-adrenergic receptor involved in bronchodilation and airway inflammation control [66]; *ADIPOQ* (*Adiponectin, C1Q And Collagen Domain Containing*), a protein with anti-inflammatory and metabolic regulatory effects [67]; *STATH* (*Statherin*), a salivary protein with antimicrobial properties that contributes to epithelial homeostasis [67]; *TNF-α* (*Tumor Necrosis Factor Alpha*), a pro-inflammatory cytokine central to immune regulation and chronic airway inflammation [68]; *TCF7L2* (*Transcription Factor 7 Like 2*), a transcription factor participating in the Wnt/β-catenin signaling pathway and glucose metabolism [69]; *IL8* (*Interleukin 8*), a chemokine that mediates neutrophil recruitment and activation in the airways [70,71]; *MUC4* and *MUC20* (*Mucin 4* and *Mucin 20*), mucin glycoproteins that contribute to mucus composition and epithelial protection [72]; *EHF* (*ETS Homologous Factor*), an epithelial transcription factor involved in cell differentiation and inflammatory regulation [72]; and *CEP72* (*Centrosomal Protein 72*), a centrosomal component critical for microtubule organization and ciliary function [72]. Although this list illustrates the diversity of modifier pathways described in CF, it is not exhaustive, and several comprehensive reviews have synthesized the expanding landscape of CF modifier genes in recent years.

This context is directly aligned with the subsequent discussion in this study, in which the rs2280091 variant of *ADAM33* was evaluated as a potential modifier gene. Pulmonary function in CF reflects a complex interplay between *CFTR* mutations, secondary modifier loci, and environmental drivers, and the associations observed in our cohort suggest that *ADAM33* variation may help explain functional heterogeneity even among patients carrying similar *CFTR* genotypes. Subsequent paragraphs expand on this relationship and position *ADAM33* within the broader mechanistic framework of airway remodeling, inflammation, and differential pulmonary function trajectories in CF.

Pulmonary function in CF is shaped by complex interactions between genetic and environmental factors. *CFTR* mutations, such as F508del, drive disease severity by promoting chronic inflammation and mucus accumulation in the airways [6,45]. Modifier genes, including *ADAM33*, may influence disease progression by regulating tissue remodeling and inflammatory processes, while environmental factors such as recurrent infections and pollutants further modulate these effects [24,25,26,44,72]. *ADAM33* has emerged as a key modifier gene implicated in airway remodeling, tissue repair, and inflammatory processes [34]. Variants in *ADAM33* have been associated with altered bronchial reactivity and progressive pulmonary function decline [34,73,74], suggesting a potential role in modulating the respiratory phenotype of patients with CF.

The *ADAM33* gene, a member of the transmembrane glycoprotein family known as A Disintegrin and Metalloproteinases (ADAMs), plays a pivotal role in biological processes such as cell adhesion, proteolysis, and extracellular matrix remodeling, making it relevant in the pathophysiology of various inflammatory and proliferative diseases [75,76]. Initially identified as an asthma susceptibility gene [76,77,78,79], *ADAM33* is expressed in vascular smooth muscle cells and tissues including the lungs, where it contributes to airway remodeling, angiogenesis, and chronic inflammation, for instance, in cases of allergic rhinitis [80,81]. Studies on nasal polyposis have shown that its upregulated expression in epithelial and mesenchymal cells suggests a pathogenic role, with potential implications in allergies and drug intolerances [78,82].

Beyond its involvement in respiratory conditions [83], *ADAM33* has been implicated in palmar dermatoglyphic patterns and cutaneous disorders such as psoriasis and atopic dermatitis, where polymorphisms consistently associated with early onset forms suggest roles in cell adhesion and epidermal remodeling [79,84,85,86,87,88]. In oncological contexts, *ADAM33* expression and differential methylation have been linked to several cancers, including breast, thyroid, and gastric malignancies. It may act as a tumor suppressor in certain settings—for instance, via truncated isoforms that inhibit the oncogenic activity of the full-length protein—or alternatively promote tumor progression by enhancing cell migration, potentially through Interleukin 18 (IL-18)–mediated mechanisms [89,90,91,92,93]. Moreover, in atherosclerosis, *ADAM33* expression in vascular lesions appears to inhibit smooth muscle cell migration, highlighting a potential protective role in vascular remodeling [94].

The rs2280091 variant of the *ADAM33* gene, although its significance varies across populations, has been consistently linked to an increased risk of allergic and functional asthma, with the G allele correlating with greater susceptibility and pulmonary function decline [74,95]. Similar observations in psoriasis suggest that this variant contributes to disease progression through shared mechanisms of inflammation and tissue remodeling, highlighting its modulatory role in systemic pathological processes [85,86,96]. This evidence supports the potential of *ADAM33*, particularly the rs2280091 variant, as a biomarker for pulmonary function progression in respiratory conditions such as asthma and CF.

The genotypic distribution among patients with CF showed a significant deviation from Hardy–Weinberg equilibrium, a phenomenon not observed in the control group. This finding suggests that the rs2280091 variant may be subject to selective pressures within the affected population. In CF, similar deviations have been described for other modifier genes whose variants can influence survival or long-term disease trajectories. For example, polymorphisms in *STAT3* (*Signal Transducer and Activator of Transcription 3*), *IL1R* (*Interleukin-1 Receptor*), and *TNFR1* (*Tumor Necrosis Factor Receptor 1*) have previously been associated with protection against chronic airway inflammation, slower functional decline, and improved prognosis, supporting the concept that modifier loci may be enriched among individuals who survive longer or experience milder disease [97,98,99].

In this context, individuals carrying the *ADAM33* rs2280091 G allele could theoretically derive functional advantages—such as better preservation of small-airway flow—leading to longer survival or greater likelihood of reaching adulthood with sufficient clinical stability to be recruited into such studies. This could contribute to the deviation from the expected genotype frequencies. Importantly, such selection would operate only within the CF population and would not influence allele distribution in the general population, given the reduced fertility and shortened lifespan typically associated with CF.

Evidence suggests that the G allele of the *ADAM33* rs2280091 variant may be associated with preserved peripheral airway function in CF, potentially through modulation of tissue remodeling, extracellular matrix integrity, and smooth muscle and fibroblast proliferation. This protective effect appears to be context-specific, contrasting with findings in asthma, where the same variant has been associated with greater susceptibility and worse pulmonary function [35,100].

This apparent paradox may reflect fundamental physiological differences between the two conditions. In asthma, airway obstruction results primarily from hyperresponsiveness, eosinophilic inflammation, and excessive tissue remodeling, and increased ADAM33 activity may exacerbate these processes, worsening airflow limitation [101]. In CF, however, lung pathology is heavily driven by neutrophilic inflammation, infection-mediated injury, and chronic structural damage rather than bronchial hyperreactivity [102,103]. In this setting, enhanced ADAM33-mediated remodeling or matrix turnover could theoretically support the structural preservation of small airways or aid tissue adaptation under chronic inflammatory stress, resulting in better spirometric performance. The distinct inflammatory microenvironments, immune pathways, and mechanisms of airway injury in asthma and CF therefore offer a plausible biological explanation for how the same *ADAM33* variant could manifest opposite clinical effects. This reinforces the importance of interpreting modifier gene effects within disease-specific context and supports the rationale for further mechanistic studies to elucidate the functional consequences of rs2280091 in CF.

In addition, in CF, chronic activation of inflammatory pathways and viscous mucus accumulation may render G allele carriers more resilient to small airway damage, reflected in superior peripheral expiratory flow parameters, including FEF25–75%, FEF50%, and MEF, without necessarily affecting bronchodilator responsiveness. These observations support *ADAM33* as a significant modifier gene in CF, influencing chronic structural and inflammatory mechanisms that contribute to the preservation of peripheral airway function.

The observed protective effect associated with the G allele of the rs2280091 variant in *ADAM33* is biologically plausible within the current understanding of the gene’s functional role in airway biology and chronic respiratory disease [101]. Although rs2280091 is a synonymous variant and therefore does not alter the amino acid sequence of the encoded protein, its position within a transcribed region may influence post-transcriptional regulatory processes, including messenger ribonucleic acid (mRNA) stability, codon usage efficiency, or RNA secondary structure, which in turn could modulate protein expression. In addition, evidence from population genomic datasets indicates that rs2280091 is in linkage disequilibrium with other *ADAM33* variants harboring recognized functional relevance, including promoter and intronic loci capable of affecting transcription factor affinity, epigenetic remodeling, and enhancer–promoter interactions. These linked variants have previously been associated with altered expression of *ADAM33* in bronchial epithelium and with downstream phenotypes such as airway hyperresponsiveness, extracellular matrix deposition, tissue remodeling, and pro-inflammatory signaling [101]. In the context of CF, where chronic infection and sustained inflammatory activation drive progressive airway damage, a reduction in ADAM33-mediated remodeling could contribute to attenuated disease severity, aligning with the protective directionality identified in the present analysis. Therefore, rather than representing an isolated synonymous effect, the G allele of rs2280091 may act as a genomic proxy for a wider regulatory haplotype that decreases ADAM33 functional impact in airway pathology. Future expression studies, chromatin accessibility assays, and haplotype-level fine-mapping analyses in CF cohorts are warranted to elucidate the mechanistic link between this variant and the respiratory outcomes observed.

### Limitations

This study has several limitations that should be considered when interpreting the results. First, the sample size was relatively small, with a predominance of young female participants, which limits the generalizability of the findings to the broader CF population. Additionally, the cross-sectional design restricts the ability to assess longitudinal progression of pulmonary function, preventing conclusions about the long-term impact of the rs2280091 variant of *ADAM33*.

The lack of stratification by ethnicity and environmental factors also limits the analysis of potential gene–environment interactions, which may significantly influence disease expression. Another important consideration is that the study focused exclusively on the rs2280091 variant, not addressing potential effects of other *ADAM33* variants or interactions with additional modifier genes that could modulate the pulmonary phenotype.

Another limitation of the present study is the limited characterization of asthma or asthma-like disease within the CF population. Because airway remodeling and hyperresponsiveness may overlap between CF and asthma, it is not possible to fully determine whether the observed associations with *ADAM33* reflect CF-specific mechanisms or concomitant airway disease resembling asthma.

A major limitation of the study is the marked predominance of female participants in the cohort, which may limit the generalizability of the findings. Although *ADAM33* is not located on a sex chromosome, sex-related differences in lung disease presentation and progression cannot be completely excluded. Future studies with more balanced sex distribution are needed to validate whether the associations observed are consistent across sexes.

Although the results indicate that carriers of the G allele, particularly GG homozygotes, exhibit relative preservation of pulmonary function, these findings should be interpreted with caution. Future longitudinal studies with larger and more heterogeneous cohorts are needed to confirm these associations and elucidate the molecular mechanisms underlying the influence of *ADAM33* on the phenotypic variability observed in CF.

## 4. Materials and Methods

### 4.1. Study Population

This was a cross-sectional observational study conducted at the Hospital das Clínicas of the University of Campinas (Universidade de Campinas—Unicamp, Campinas, São Paulo, Brazil) and at the University of São Francisco (Universidade São Francisco—USF, Bragança Paulista, São Paulo, Brazil). The study protocol was approved by the Research Ethics Committee [Certificate of Presentation for Ethical Consideration (Certificado de Apresentação para Apreciação Ética—CAAE) no. 38162914.3.0000.5404], and all participants—or their legal guardians—provided written informed consent, in accordance with the Declaration of Helsinki.

A total of 55 patients CF were included. The diagnosis was established based on an abnormal sweat test (chloride ion concentration > 60 mEq/L) and confirmation of two pathogenic variants in the *CFTR* gene. Inclusion criteria comprised age ≥ 6 years, ability to perform reliable spirometry, and clinical stability, defined as the absence of pulmonary exacerbation in the preceding four weeks. Patients with significant comorbidities that could interfere with lung function were excluded.

For clinical characterization, demographic and anthropometric data were collected, including age (years), sex (male or female), ethnicity (White, Black, Mixed-race, or Asian), weight (kg), height (m), and BMI (calculated as kg/m^2^). Additional data included history of respiratory exacerbations and spirometric measurements performed before and after bronchodilator administration, following the American Thoracic Society/European Respiratory Society (ATS/ERS) recommendations [104].

As a control group, allele frequencies were obtained from the ABraOM database, which includes genomic data from 608 elderly (> 60 years), genetically admixed individuals residing in São Paulo, representing the genetic diversity of the Brazilian population, with predominant African, European, and Indigenous ancestry [40]. ABraOM participants were healthy elderly individuals without a history of severe disease, providing an appropriate reference population for genetic comparison [40]. The use of ABraOM as a control enables population-based comparison with a non-affected cohort, offering reference values for the rs2280091 variant frequency in the *ADAM33* gene [40].

### 4.2. Pulmonary Function

Pulmonary function was assessed by spirometry before and after bronchodilator administration, strictly following the ATS/ERS (2019) standards [104]. The following clinical markers of pulmonary function were analyzed:

1. FVC: the total volume of air exhaled with maximal effort, starting from full inspiration. It reflects the individual’s pulmonary reserve and is essential for assessing ventilatory function.

2. FEV1: the volume of air exhaled during the one second of the FVC maneuver, serving as a sensitive marker of airway obstruction.

3. FEV1/FVC ratio: expresses the proportion of air exhaled in the first second relative to the total exhaled volume; reduced values indicate airflow obstruction.

4. Forced Expiratory Flow (FEF) at 25%, 50%, and 75% of FVC (FEF25%, FEF50%, and FEF75%): reflects the mean expiratory flow at different points during exhalation, providing information about small and large airways.

5. FEF25–75%: represents the average expiratory flow between 25% and 75% of FVC, a sensitive marker of medium and small airway involvement, frequently compromised in CF.

6. Maximal Expiratory Forced flow (MEF): the highest airflow achieved during the FVC maneuver, reflecting overall airway function and serving as a sensitive marker for early airflow obstruction.

These parameters provide a comprehensive assessment of pulmonary function, enabling accurate characterization of obstruction severity and ventilatory heterogeneity among the patients studied. All spirometric values were expressed as percentages of predicted values, based on the Global Lung Function Initiative (GLI-2012) reference equations [105].

### 4.3. Analysis of the rs2280091 Variant in the ADAM33 Gene

Genomic DNA was extracted from peripheral blood samples previously stored in a biorepository, following standardized preservation protocols. The single nucleotide variant (SNV) rs2280091 of the *ADAM33* gene was genotyped using the Polymerase Chain Reaction-Restriction Fragment Length Polymorphism (PCR-RFLP) method. The resulting fragments were visualized on 2% agarose gels stained with SYBR^®^ Safe DNA Gel Stain (1X) (ThermoFisher Scientific, Waltham, MA, USA). The technical details of the reaction, including primer sequences and enzyme digestion conditions, are described below.

For DNA amplification, all samples were previously quantified to ensure optimal concentrations between 50 and 500 ng/µL, thereby guaranteeing PCR efficiency. The reaction conditions were optimized for a final volume of 25 µL, containing 1 µL of genomic DNA, 0.5 µL of each primer (5 pmol), 0.75 U of Uniscience DNA Polymerase (5 U/µL), 2 mM MgCl_2_, 50 µM of each deoxyribonucleotide, and 1X reaction buffer containing KCl (50 mM KCl; 75 mM Tris-HCl, pH 9.0; 20 mM (NH_4_)_2_SO_4_). All reactions were performed in parallel with a negative (blank) control.

The amplification was carried out using a touchdown PCR protocol, as outlined in Table 6.

Following amplification, the resulting fragments were subjected to enzymatic digestion for approximately 16 h using a PTC-100™ Programmable Thermal Controller (MJ Research Inc., Saint-Bruno-de-Montarville, QC, Canada). For the rs2280091 variant, the digestion reaction had a final volume of 15 µL, consisting of 10 µL of amplified DNA, 5 U of the NcoI restriction enzyme (10 U/µL), and 1X ThermoFisher Scientific Tango Buffer™, (ThermoFisher Scientific, Waltham, MA, USA) incubated at 37 °C.

The primer sequences, the restriction enzyme used for cleavage of the amplified DNA, and the resulting fragment sizes are summarized in Table 7. For genotyping of the rs2280091 variant in the *ADAM33* gene, the following primers were used: T1 A/G F: 5′-ACTCAAGGTGACTGGGTGCT-3′ and T1 A/G R: 5′-GAGGGCATGAGGCTCACTTG-3′.

Enzymatic digestion with NcoI produced fragments of different lengths depending on the allele present: the A allele generated 140 bases pairs (bp) and 260 bp fragments, whereas the G allele produced a single 400 bp fragment.

The digested products were analyzed by 2% agarose gel electrophoresis, stained with SYBR^®^ Safe DNA Gel Stain (1X), and separated according to molecular size, allowing for unequivocal allele identification. This approach ensured high sensitivity and specificity for detecting the *ADAM33* rs2280091 variant, enabling robust association analyses with clinical and spirometric parameters.

### 4.4. Statistical Analysis

Genotypic frequencies were assessed for Hardy–Weinberg equilibrium using the chi-square (χ^2^) test implemented in the OEGE. *p*-values < 0.05 were considered indicative of deviation from equilibrium.

Allelic and genotypic frequencies of patients with CF were compared with those of the reference population from ABraOM using the χ^2^ test.

Spirometric parameters, including FVC, FEV1, FEF25%, FEF50%, FEF75%, MEF, and FEF25–75%, were analyzed using the Kruskal–Wallis and Mann–Whitney tests for independent samples. Numerical variables were presented as median with interquartile range (25th–75th percentiles), whereas categorical variables were expressed as absolute frequencies and percentages. Additionally, ORs and corresponding 95% CI were calculated using OpenEpi software (version 3.01) [42].

All statistical analyses were performed using the Statistical Package for the Social Sciences (SPSS) software (IBM SPSS Statistics for Macintosh, Version 28.0). Graphical representations and figures were generated using GraphPad Prism version 10.2.3 (GraphPad Software, Boston, MA, USA; www.graphpad.com).

A significance level of 5% (*p*-values < 0.05) was adopted for all analyses. In this study, the FDR was controlled to reduce the likelihood of type I errors arising from multiple statistical comparisons. When a large number of hypotheses are tested simultaneously, the probability of obtaining statistically significant results by chance alone increases. To address this, we applied the Benjamini–Hochberg procedure, a widely accepted method for FDR correction, as implemented by the FDR calculator platform. This approach ranks the individual *p*-values from lowest to highest and determines the largest *p*-value that satisfies the threshold for statistical significance at a predefined FDR level. All values equal to or smaller than this cutoff are then considered significant after correction. Unlike more conservative methods such as Bonferroni adjustment, the Benjamini–Hochberg procedure maintains greater statistical power while still controlling the expected proportion of false positives among the results deemed significant.

In the present dataset, FDR correction was applied over three simultaneous comparisons, corresponding to the analytical groups presented in the title of Table 4: (i) genotype comparisons (AA vs. AG vs. GG), (ii) grouped genotypes (AA vs. AG + GG), and (iii) allelic comparisons (A vs. G) for the rs2280091 variant of the *ADAM33* gene in individuals with CF. Therefore, the corrected *p*-values reflect the adjustment for these three levels of analysis, ensuring that the reported associations remain robust after controlling for multiple testing. The application and rationale of this correction are explicitly described in Section 4 and indicated in Section 2 to ensure transparency and reproducibility.

The primary outcome of the study was to evaluate the association between the rs2280091 genotype of the *ADAM33* gene and spirometric parameters. The secondary outcome was to compare the allelic and genotypic frequencies observed in CF patients with those reported for the Brazilian reference population from ABraOM.

## 5. Conclusions

Our findings indicate that the *ADAM33* rs2280091 G allele is linked to enhanced spirometric performance in CF patients, particularly in parameters reflecting small and medium airway flows. Rather than suggesting a direct association with increased disease risk, these results may reflect a survivor effect, in which carriers of the G allele are more likely to reach clinical follow-up and study recruitment. This potential biological advantage underscores the role of *ADAM33* as a modifier gene influencing disease phenotype and therapeutic responsiveness in CF, reinforcing the value of integrating genetic profiling with pulmonary functional assessment for personalized management and risk stratification in CF.

## Figures and Tables

**Figure 1 ijms-26-11583-f001:**
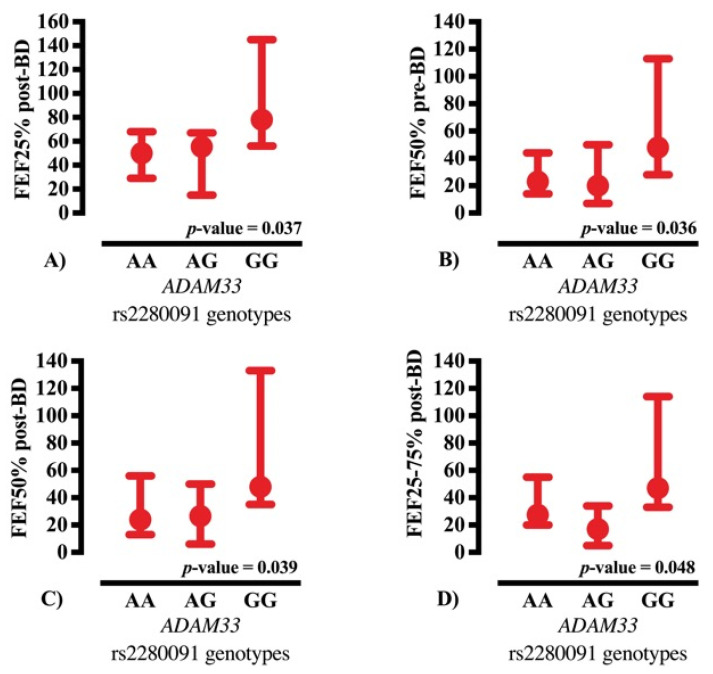
Comparison of major spirometric parameters among cystic fibrosis patients according to *ADAM33* rs2280091 genotypes. Data are presented as median and 95% confidence interval (95% CI). Panels represent: (**A**) FEF25% (% of predicted) post-bronchodilator; (**B**) FEF50% (% of predicted) pre-bronchodilator; (**C**) FEF50% (% of predicted) post-bronchodilator; and (**D**) FEF25–75% (% of predicted) post-bronchodilator. Statistical analyses were performed using non-parametric tests, and *p*-values were corrected for multiple testing using the Benjamini–Hochberg false discovery rate (FDR) method (corrected *p*-value). %: percentage, *ADAM33*: *ADAM metallopeptidase domain 33*, BD: bronchodilator, FEF25%: Forced Expiratory Flow at 25% of FVC, FEF25–75%: Forced Expiratory Flow between 25% and 75% of FVC, FEF50%: Forced Expiratory Flow at 50% of FVC, FVC: Forced Vital Capacity.

**Figure 2 ijms-26-11583-f002:**
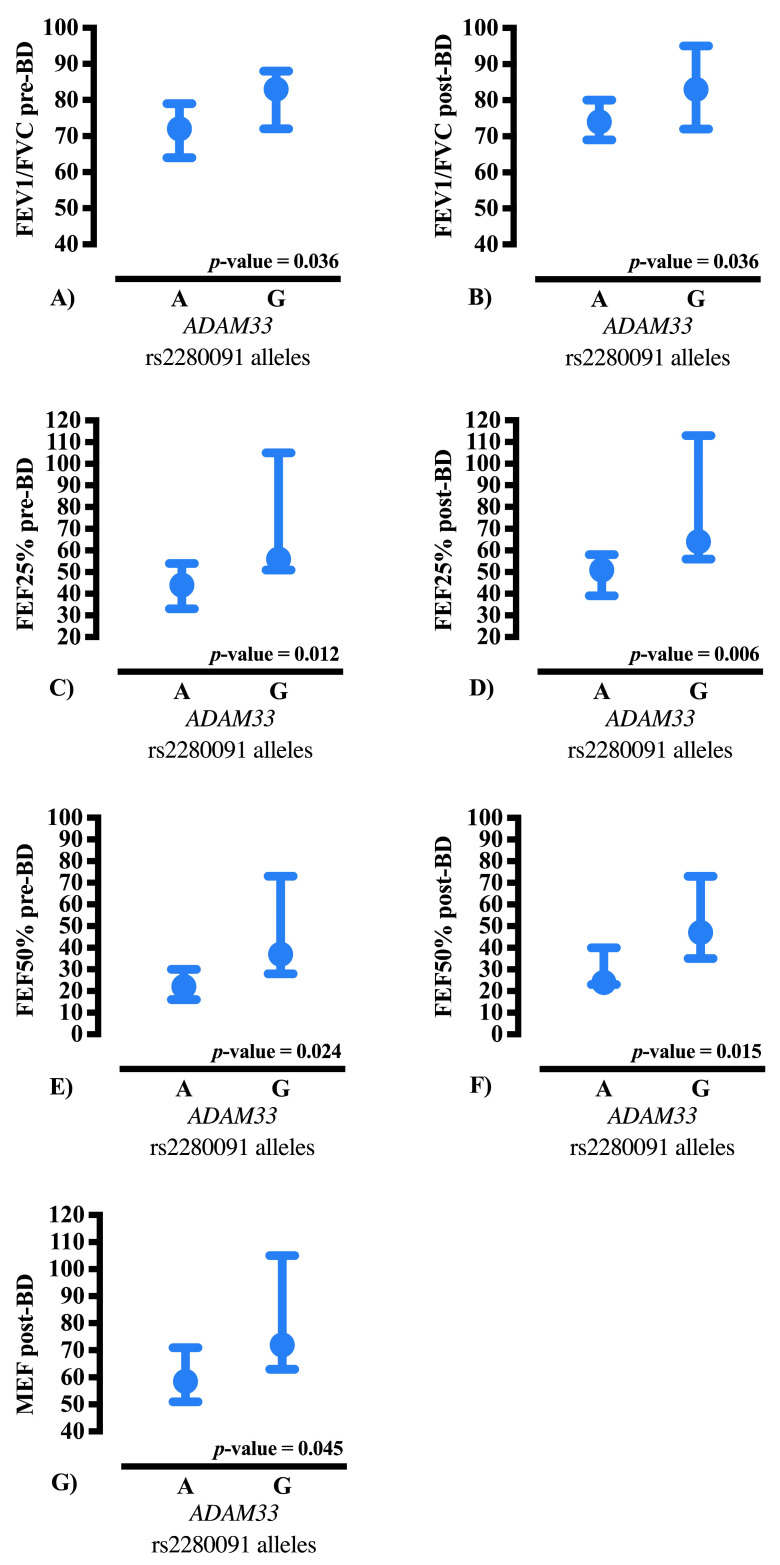
Comparison of major spirometric parameters among cystic fibrosis patients according to *ADAM33* rs2280091 alleles. Data are presented as median and 95% confidence interval (95% CI). Panels represent: (**A**) FEV1/FVC ratio (% of predicted) pre-bronchodilator; (**B**) FEV1/FVC ratio (% of predicted) post-bronchodilator; (**C**) FEF25% (% of predicted) pre-bronchodilator; (**D**) FEF25% (% of predicted) post-bronchodilator; (**E**) FEF50% (% of predicted) pre-bronchodilator; (**F**) FEF50% (% of predicted) post-bronchodilator; and (**G**) MEF (% of predicted) post-bronchodilator. Comparisons were performed using non-parametric statistical methods (Mann–Whitney U test for independent samples), and all *p*-values shown correspond to multiple testing correction using the Benjamini–Hochberg false discovery rate (FDR) procedure (corrected *p*-value). %: percentage, *ADAM33*: *ADAM metallopeptidase domain 33*, FEF25%: Forced Expiratory Flow at 25% of FVC, FEF50%: Forced Expiratory Flow at 50% of FVC, FEV1: Forced Expiratory Volume in one second of FVC, FVC: Forced Vital Capacity, MEF: Maximal Expiratory Forced flow.

**Table 1 ijms-26-11583-t001:** Descriptive analysis of study participants according to sex, ethnicity, age, anthropometric data, and *CFTR* gene variants.

Marker	Distribution
Sex (female)	53 (96.4%)
Self-declared Caucasian ethnicity	29 (52.7%)
Age (years)	17 (14–21)
Weight (kg)	48.70 (36.70–54.60)
Height (m)	1.59 (1.51–1.64)
Body mass index (kg/m^2^)	17.89 (16.66–20.36)
***CFTR* gene variants**	
F508del/F508del	29 (52.7%)
F508del/G542*	9 (16.4%)
F508del/1812-1G>A	3 (5.5%)
F508del/N1303K	3 (5.5%)
F508del/1717-1G>A	2 (3.6%)
F508del/R1162*	2 (3.6%)
F508del/R553*	1 (1.8%)
F508del/G85E	1 (1.8%)
F508del/P205S	1 (1.8%)
F508del/2183AA>G	1 (1.8%)
A561E/A561E	1 (1.8%)
3120+1G>A/R1066C	1 (1.8%)
2183AA>G/2183AA>G	1 (1.8%)

*CFTR*: cystic fibrosis transmembrane conductance regulator, kg: kilograms, m: meters. Data are presented as absolute (relative) frequency or as median (25th to 75th percentile).

**Table 2 ijms-26-11583-t002:** Comprehensive description of *CFTR* gene variants identified in cystic fibrosis participants, considering allelic distribution.

Traditional Name	N	%	cDNA	Protein	CFTR Functional Class	dbSNP ID	CFF (%) ^a^	Brazil (%) ^b^
G542*	9	8.18	c.1624G>T	p.Gly542Ter	IB	rs113993959	2.72	6.94
R1162*	2	1.82	c.3484C>T	p.Arg1162Ter	IB	rs74767530	0.5	1.99
2183AA>G	3	2.73	c.2051_2052delAAinsG	p.Lys684SerfsX38	IA	rs121908799	0.45	0.21
1717-1G>A	2	1.82	c.1585-1G>A	Not applicable	IA	rs76713772	0.82	0.34
3120+1G>A	1	0.91	c.2988+1G>A	Not applicable	IA	rs75096551	0.46	3.02
1812-1G>A	3	2.73	c.1680-1G>A	Not applicable	IA	rs121908794	0.04	0.40
R553*	1	0.91	c.1657C>T	p.Arg553Ter	IB	rs74597325	0.89	0.36
F508del	81	73.64	c.1521_1523delCTT	p.Phe508del	II	rs113993960	65.07	44.39
N1303K	3	2.73	c.3909C>G	p.Asn1303Lys	II	rs80034486	1.68	1.11
A561E	2	1.82	c.1682C>A	p.Ala561Glu	II	rs121909047	0.04	0.72
R1066C	1	0.91	c.3196C>T	p.Arg1066Cys	II	rs78194216	0.25	1.28
G85E	1	0.91	c.254G>A	p.Gly85Glu	II	rs75961395	0.52	1.57
P205S	1	0.91	c.613C>T	p.Pro205Ser	IV	rs121908803	0.06	0.68

%: percentage, cDNA: complementary Deoxyribonucleic Acid, CFF: Cystic Fibrosis Foundation, *CFTR*: cystic fibrosis transmembrane conductance regulator, dbSNP ID: Single Nucleotide Polymorphism database identifier, N: number of alleles. Notes: ^a^ Based on the CFTR2 patient registry (9 September 2024), which includes 122,935 patients and 1167 variants: 1085 disease-causing, 55 of variable clinical consequence, and 27 non–disease-causing. ^b^ Based on the Brazilian Cystic Fibrosis Registry (Registro Brasileiro de Fibrose Cística—REBRAFC) including 1760 patients.

**Table 3 ijms-26-11583-t003:** Descriptive analysis of the rs2280091 variant of the *ADAM33* gene in participants with cystic fibrosis.

Genotype	N (%)—Cystic Fibrosis	N (%)—Healthy Control ^a^	Odds Ratio (95% CI)	*p*-Value (Adjusted *p*-Value) *,**
AA	38 (69.1%)	478 (78.62%)	0.608 (0.332–1.112)	0.103 (0.129)
AG	7 (12.7%)	119 (19.57%)	0.559 (0.265–1.358)	0.216 (0.216)
GG	10 (18.2%)	11 (0.02%)	**12.06 (4.863–29.91)**	**<0.001 (0.003)**
**Genotypic grouping**				
AA	38 (69.1%)	478 (78.62%)	0.608 (0.332–1.112)	0.103 (0.129)
AG + GG	17 (30.9%)	130 (21.38%)	Reference	
**Allelic frequency**				
A	83 (75.5%)	1075 (88.4%)	**0.403 (0.252–0.644)**	**<0.001 (0.003)**
G	27 (24.5%)	141 (11.59%)	Reference	

95% CI: 95% confidence interval, %: percentage, *ADAM33*: *ADAM metallopeptidase domain 33*, N: number of participants. *: Statistical analysis was performed using the χ^2^ test, **: *p*-values were corrected for multiple testing using the Benjamini–Hochberg false discovery rate method. Hardy–Weinberg equilibrium (HWE) was assessed using the OEGE platform (Online Encyclopedia for Genetic Epidemiology Studies) ^b^ with the χ^2^ test. A *p*-value below 0.05 indicates deviation from HWE. In this study, the rs2280091 variant deviated from equilibrium in the cystic fibrosis group but remained in equilibrium in the healthy control group. All analyses were conducted with a significance level (α) of 0.05. Statistically significant results (*p* < 0.05) are presented in bold. Odds ratios were calculated using the OpenEpi platform ^c^ (version 3.01), employing the Taylor series method for 95% CI estimation. ^a^ Naslavsky et al. 2017 [40], ^b^ Rodriguez et al. 2009 [41], ^c^ Sullivan et al. 2009 [42].

**Table 4 ijms-26-11583-t004:** Association [*p*-value; false discovery rate (FDR)-adjusted *p*-value] * between clinical and functional markers according to genotypes (AA vs. AG vs. GG), grouped genotypes (AA vs. AG + GG), and alleles (A vs. G) for the rs2280091 variant of the *ADAM33* gene in cystic fibrosis participants.

Marker	AA vs. AG vs. GG ^a^	AA vs. AG + GG ^b^	A vs. G ^b^
Age (years)	0.445 (0.818)	0.784 (0.818)	0.818 (0.818)
Weight (kg)	0.781 (0.781)	0.506 (0.759)	0.483 (0.759)
Height (m)	0.630 (0.630)	0.438 (0.630)	0.232 (0.630)
Body mass index (kg/m^2^)	0.880 (0.880)	0.623 (0.880)	0.588 (0.880)
FVC (%)	0.210 (0.315)	0.669 (0.669)	0.196 (0.315)
FVC bronchodilator response	0.993 (0.993)	0.916 (0.993)	0.879 (0.993)
FVC (%) post-BD	0.098 (0.147)	0.535 (0.535)	0.091 (0.147)
FEV1 (%)	0.127 (0.191)	0.455 (0.455)	0.079 (0.191)
FEV1 bronchodilator response	0.714 (0.714)	0.496 (0.714)	0.301 (0.714)
FEV1 (%) post-BD	0.071 (0.108)	0.512 (0.512)	0.072 (0.108)
FEV1/FVC (%)	0.058 (0.087)	0.177 (0.177)	**0.012 (0.036)**
FEV1/FVC bronchodilator response	0.963 (0.963)	0.796 (0.963)	0.721 (0.963)
FEV1/FVC (%) post-BD	0.117 (0.117)	0.114 (0.117)	**0.012 (0.036)**
FEF25% (%)	**0.046** (0.069)	0.097 (0.097)	**0.004 (0.012)**
FEF25% bronchodilator response	0.695 (0.888)	0.888 (0.888)	0.581 (0.888)
FEF25% (%) post-BD	**0.025 (0.037)**	0.075 (0.075)	**0.002 (0.006)**
FEF50% (%)	**0.024 (0.036)**	0.193 (0.193)	**0.008 (0.024)**
FEF50% bronchodilator response	0.701 (0.701)	0.566 (0.701)	0.676 (0.701)
FEF50% (%) post-BD	**0.026 (0.039)**	0.136 (0.136)	**0.005 (0.015)**
FEF75% (%)	0.096 (0.231)	0.702 (0.702)	0.154 (0.231)
FEF75% bronchodilator response	0.916 (0.916)	0.810 (0.916)	0.891 (0.916)
FEF75% (%) post-BD	0.064 (0.096)	0.470 (0.470)	0.059 (0.096)
FEF25–75% (%)	**0.046** (0.090)	0.506 (0.506)	0.060 (0.090)
FEF25–75% bronchodilator response	0.675 (0.675)	0.477 (0.675)	0.535 (0.675)
FEF25–75% (%) post-BD	**0.016 (0.048)**	0.673 (0.673)	0.071 (0.107)
MEF (%)	**0.029** (0.084)	0.542 (0.084)	0.056 (0.542)
MEF bronchodilator response	0.570 (0.590)	0.590 (0.590)	0.285 (0.590)
MEF (%) post-BD	**0.037** (0.055)	0.251 (0.251)	**0.015 (0.045)**

%: percentage, *ADAM33*: *ADAM metallopeptidase domain 33*, BD: bronchodilator, FEF25%: Forced Expiratory Flow at 25% of FVC, FEF25–75%: mean Forced Expiratory Flow between 25% and 75% of FVC, FEF50%: Forced Expiratory Flow at 50% of FVC, FEF75%: Forced Expiratory Flow at 75% of FVC, FEV1: Forced Expiratory Volume in one second, FVC: Forced Vital Capacity, kg: kilograms, m: meters, MEF: Maximal Expiratory Forced flow. ^a^ Statistical analysis performed using the Kruskal–Wallis test for independent samples. ^b^ Statistical analysis performed using the Mann–Whitney U test for independent samples. All analyses were conducted with a significance level (α) of 0.05. Statistically significant results (*p*-value < 0.05) are presented in bold. * *p*-values were corrected for multiple testing using the Benjamini–Hochberg FDR method.

**Table 5 ijms-26-11583-t005:** Comparison between genotypes and alleles for spirometry parameters showing significant association with the rs2280091 variant of the *ADAM33* gene in participants with cystic fibrosis.

**Parameter (% Predicted) ^a^**	**AA**	**AG**	**GG**	***p*-Value (FDR-Adjusted *p*-Value) ***
FEF25% pre-bronchodilator	38; 43 (22.25–76)	7; 51 (15–56)	10; 82 (50–117.25)	**0.046** (0.069)
FEF25% post-bronchodilator	31; 50 (21–80)	6; 55.50 (18.75–62.50)	9; 78 (56.50–137.50)	**0.025 (0.037)**
FEF50% pre-bronchodilator	38; 23 (9.50–53.25)	7; 20 (8–35)	10; 48 (28–84.50)	**0.024 (0.036)**
FEF50% post-bronchodilator	31; 24 (12–62)	6; 26.50 (10.50–41.75)	9; 48 (41–121)	**0.026 (0.039)**
FEF25–75% pre-bronchodilator	38; 23.50 (10.75–55.75)	7; 17 (11–29)	10; 38.50 (27–79.25)	**0.046** (0.090)
FEF25–75% post-bronchodilator	30; 27.5 (12.75–64)	6; 17 (6.5–24.25)	9; 47 (35–110)	**0.016 (0.048)**
MEF pre-bronchodilator	38; 60.50 (47.50–89.75)	7; 51 (43–57)	10; 82.50 (56–117.25)	**0.029** (0.084)
MEF post-bronchodilator	31; 61 (45–85)	6; 55 (45–85)	9; 84 (66–124.50)	**0.037** (0.055)
**Parameter (% Predicted) ^b^**	**Allele A**		**Allele G**	
FEV1/FVC (%) pre-bronchodilator	83; 72 (62–83)		27; 83 (72–88)	**0.012 (0.036)**
FEV1/FVC (%) post-bronchodilator	68; 74 (65–82.75)		24; 83 (72–96.50)	**0.012 (0.036)**
FEF25% pre-bronchodilator	83; 44 (23–75)		27; 56 (47–112)	**0.004 (0.012)**
FEF25% post-bronchodilator	68; 51 (21–75)		24; 64 (56–125.75)	**0.002 (0.006)**
FEF50% pre-bronchodilator	83; 22 (9–50)		27; 37 (28–73)	**0.008 (0.024)**
FEF50% post-bronchodilator	68; 24 (12–57)		24; 47 (32–100)	**0.005 (0.015)**
MEF post-bronchodilator	68; 58.50 (45.50–79)		24; 72 (57.75–113.25)	**0.015 (0.045)**

%: percentage, *ADAM33: ADAM metallopeptidase domain 33*, FDR: false discovery rate, FEF25%: Forced Expiratory Flow at 25% of FVC, FEF25–75%: Forced Expiratory Flow between 25% and 75% of FVC, FEF50%: Forced Expiratory Flow at 50% of FVC, FEV1: Forced Expiratory Volume in one second of FVC, FVC: Forced Vital Capacity, MEF: Maximal Expiratory Forced flow. ^a^ Statistical analysis performed using the Kruskal–Wallis test for independent samples. ^b^ Statistical analysis performed using the Mann–Whitney U test for independent samples. The data are presented as number of patients; median with interquartile range (25th–75th percentiles). All analyses were conducted with a significance level (α) of 0.05. Statistically significant results (*p*-values < 0.05) are presented in bold. * *p*-values were corrected for multiple testing using the Benjamini–Hochberg FDR method.

**Table 6 ijms-26-11583-t006:** Touchdown polymerase chain reaction (PCR) conditions for genotyping of the rs2280091 variant in the *ADAM33* gene.

Step	Temperature	Time	Cycles
Initial denaturation	94 °C	5 min	–
Touchdown cycle	94 °C	1 min	20 cycles
Annealing	65.5 °C	1 min	Decrease of 0.5 °C per cycle (20 cycles)
Extension	72 °C	2 min	20 cycles
Standard cycle	94 °C	1 min	30 cycles
Annealing	55.5 °C	1 min	30 cycles
Extension	72 °C	2 min	30 cycles
Final extension	72 °C	10 min	–

°C: degrees Celsius, ADAM33: A Disintegrin and Metallopeptidase Domain 33, min: minutes.

**Table 7 ijms-26-11583-t007:** Primers, restriction enzyme, and expected fragment sizes after digestion for genotyping of the rs2280091 variant in the *ADAM33* gene.

Variant	Primer	Sequence (5′→3′)	Restriction Enzyme *	Expected Fragments After Digestion
rs2280091	T1 A/G F	ACTCAAGGTGACTGGGTGCT	NcoI	Allele A: 140 and 260 bp
	T1 A/G R	GAGGGCATGAGGCTCACTTG		Allele G: 400 bp

*ADAM33*: *A Disintegrin and Metallopeptidase Domain 33*, bp: base pairs, NcoI: type II restriction endonuclease derived from *Nocardia corallina*, Primer F (Forward): oligonucleotide primer in the 5′→3′ direction for DNA (deoxyribonucleic acid) amplification, Primer R (Reverse): complementary oligonucleotide primer in the opposite direction (3′→5′). *, The NcoI restriction enzyme cleaves only the A allele due to the presence of the recognition sequence (CC*ATGG), while the G allele remains uncut.

## Data Availability

The original contributions presented in this study are included in the article/Appendix A. Further inquiries can be directed to the corresponding author.

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
