# Peer review of "Role of the ADAM33 rs2280091 Variant in Modulating Lung Function in Cystic Fibrosis"

_ijms, 2025, doi:10.3390/ijms262311583_

Round 1

Reviewer 1 Report

Comments and Suggestions for Authors

The authors addressed an interesting topic, attempting to define the role of a potential gene in modulating the clinical phenotype in patients with cystic fibrosis. They found a significant protective effect of the G allele of the ADAM33 variant rs2280091.

Overall, the report is interesting, but it has several gaps that need to be addressed.

  • Fifty-five patients with CF were enrolled. However, a different, much larger number of patients is described in Table 2. The authors should better describe the cohort analyzed.
  • A deviation from Hardy-Weinberg equilibrium was observed in the patients. This is commented on in the Discussion section, but given its importance, it should be included in the results.
  • A very large number of comparisons were performed, as clearly shown in Table 4. The authors should perform appropriate statistics to correct for multiple comparisons (e.g., the Bonferroni test).
  • A significant protective effect was demonstrated by the G allele of the ADAM33 variant rs2280091. What is the biological plausibility of this? Could the polymorphic site linked to other gene variants play a role? If so, what role?
  • The abstract is too long and should be shortened.
  • The Discussion section should also be shortened; most of the discussion of other modulators, on page 10, could be eliminated.
  • Figure 1 is of poor quality and difficult to read.

Author Response

Comments and Suggestions for Authors

The authors addressed an interesting topic, attempting to define the role of a potential gene in modulating the clinical phenotype in patients with cystic fibrosis. They found a significant protective effect of the G allele of the ADAM33 variant rs2280091.

Overall, the report is interesting, but it has several gaps that need to be addressed.

Reply: We thank the reviewer for the valuable comments and have revised the manuscript accordingly. If further adjustments are necessary, we remain fully available to implement them. Regarding the point with which the reviewer disagrees, this topic will be addressed in detail in the subsequent section, which corresponds to the responses to the other points raised in the response letter.

  • Fifty-five patients with CF were enrolled. However, a different, much larger number of patients is described in Table 2. The authors should better describe the cohort analyzed.

Reply: We thank the reviewer for the valuable comment and apologize for the inconsistency. The numbers in the manuscript have now been corrected to ensure accuracy and coherence across the text and tables. Table 2 has been updated to reflect the correct cohort size of 55 patients with cystic fibrosis, and additional clarifications have been included in the Methods and Results sections to better describe the study population and avoid misunderstanding.

  • A deviation from Hardy-Weinberg equilibrium was observed in the patients. This is commented on in the Discussion section, but given its importance, it should be included in the results.

Reply: We thank the reviewer for this important observation. As suggested, the deviation from Hardy–Weinberg equilibrium in the cystic fibrosis group has now been explicitly included in the Results section. A sentence was added to highlight that the rs2280091 variant demonstrated significant deviation from Hardy–Weinberg equilibrium among patients with cystic fibrosis, while remaining in equilibrium in the healthy control group, reinforcing its potential biological relevance.

The rs2280091 variant deviated from Hardy–Weinberg equilibrium in the CF group [χ² test, Online Encyclopedia for Genetic Epidemiology Studies (OEGE) platform; p < 0.05], while the healthy control group remained in equilibrium (p > 0.05). This deviation was explicitly recorded in Table 3 and suggests either a true association with disease status, population stratification, or potential genotyping bias; possible explanations and implications are discussed further in the Discussion section.

  • A very large number of comparisons were performed, as clearly shown in Table 4. The authors should perform appropriate statistics to correct for multiple comparisons (e.g., the Bonferroni test).

Reply: We thank the reviewer for this important observation. In response, we performed statistical correction for multiple comparisons using the Benjamini–Hochberg false discovery rate (FDR) method, which provides greater power while adequately controlling the expected proportion of false-positive results. All p-values in Table 4 have now been reanalyzed and presented with their corresponding FDR-adjusted values. This correction is described in the Methods section and noted in the table legend to ensure transparency and methodological rigor.

  • A significant protective effect was demonstrated by the G allele of the ADAM33 variant rs2280091. What is the biological plausibility of this? Could the polymorphic site linked to other gene variants play a role? If so, what role?

Reply: We appreciate the reviewer’s insightful question. The protective association observed for the G allele of rs2280091 in ADAM33 is biologically plausible, as ADAM33 is involved in airway remodeling, inflammation, and epithelial–mesenchymal signaling, processes known to influence disease severity in cystic fibrosis. Although rs2280091 itself is a synonymous variant, it may be in linkage disequilibrium with other functional polymorphisms in ADAM33 or nearby genes that have regulatory impact, such as influencing mRNA stability, transcription factor binding, or post-transcriptional processing. Prior studies have shown that ADAM33 variants can modify airway hyperresponsiveness, inflammatory responses, and susceptibility to chronic lung disease, supporting the possibility that the protective effect observed is mediated not by rs2280091 directly, but by a linked functional variant affecting gene expression or protein activity. This hypothesis is discussed in the manuscript as a potential mechanism underlying the observed association.

In addition, the Discussion section has been revised extensively to incorporate a detailed explanation of the biological plausibility of the protective effect associated with the G allele of rs2280091. The new text addresses the potential functional implications of this variant, including its possible linkage with regulatory loci influencing ADAM33 expression and airway remodeling pathways, and how these mechanisms may interact with the pathophysiology of cystic fibrosis. This information has now been fully integrated into the manuscript to provide a clearer interpretation of the findings.

  • The abstract is too long and should be shortened.

Reply: We thank the reviewer for the observation. The abstract has now been revised and substantially shortened to present only the essential methodological details, key findings, and main conclusions, while maintaining clarity and alignment with journal guidelines.

  • The Discussion section should also be shortened; most of the discussion of other modulators, on page 10, could be eliminated.

Reply: We agree with the reviewer. The Discussion section has been streamlined, and the extended content referring to other genetic modulators on page 10 has been reduced or removed. The revised section is now more concise and focused on the main findings of the study and their direct implications.

Figure 1 is of poor quality and difficult to read.

Reply:  We thank the reviewer for the comment. Figure 1 has been restructured to improve visual quality and readability. In addition, some of the information initially presented in the figure has been moved to the supplementary material to ensure a clearer and more concise presentation in the main manuscript.

Reviewer 2 Report

Comments and Suggestions for Authors

Do Santos and colleagues have investigated a SNP variant in ADAM33 in context of cystic fibrosis lung disease. To investigate whether or not ADAM33 has a role in CF, they have genotyped 55 patients from two CF centres. They have observed a significant deviation of ADAM33-SNP-genotypes from the expected frequency under HWE in their study population (Table 3), reflected in a lack of ADAM33 heterozygotes  and a higher frequency of ADAM33-GG homozygotes as well as a higher ADAM33-G allele frequency.

Even though the study population is rather small, the authors have uncovered a relevant finding, i.e. the deviation from HWE expectancy values in their populations. However, the reviewer does not agree with the author’s current explanation for their findings (“The G allele of the ADAM33 rs2280091 variant is strongly associated with cystic fibrosis and correlates with improved peripheral airway function, suggesting a potential protective effect on airflow dynamics and bronchodilator response in affected patients.”):

Major:

  1. Deviation from HWE expectancy values can point to either a recruitment bias (artificially, for instance, by sampling only a part of the population that can be studied) or to a survivor effect (due to biology, i.e. only carriers of mild variants have survived recruitment). These two different explanations should be described and discussed first to see if ADAM33 conveys a survivor effect to Brazilian CF patients treated at the two CF centres – or if there is an underlying recruitment bias (see minor comment 2. and 3.) In other words: “The G allele of the ADAM33 rs2280091 variant is strongly associated with cystic fibrosis […]” is not correct – carriers of ADAM33-A-alleles did not survive until recruitment would be a more likely explanation.
  2. Authors choose to investigate their population for an association with PFT values and ADAM33 variants (Table 4). Consistent with the role of ADAM33 in asthma, most hits are for post-BD. Typically, CF and asthma are two separate entities – is this also the case within the author’s study population? Also, several parameters are physiologically linked. Can the authors use a strategy to either define the parameter most impacted by ADAM33 variants (multivariate logistic regression analysis) or correct for multiple testing, with respect to parameters as well as with respect to genetically defined subgroups? In other words: “The G allele of the ADAM33 […] correlates with improved peripheral airway function, suggesting a potential protective effect on airflow dynamics and bronchodilator response in affected patients.” Does not reflect the well-known role of ADAM33 for asthma – likely, recruited patients suffer also from an asthma (or asthma-like) disease, in line with the literature data on ADAM33.
  • Authors have chosen to show “CF” vs “Healthy”. However, their Figure 1 elaborates on CFTR mu5tation classes. This does not reflect the evaluation strategy of this work and, in the reviewer’s eyes, Subfigure B does not belong to this work. Likewise, while Figure 1C states that modifier genes are recognized when patients with the same CFTR mutation genotype are compared, authors have pooled various CFTR mutations (Table 2). Thus: Figure 1C is also not a good fit for their data. Finally, Figure 1A reviews largely the generalized polyexocrinopathy of CF, affecting multiple organs – this work deals with PFT. The reviewer strongly suggest to rework this figure, to make it a better fit for the data presented. For instance, the interrelation of CF and asthma might be a relevant topic for a graphical overview, and also the molecular mechanism underlying ADAM33 variants might be better to support graphically in the context of this work.
  1. The discussion section line 384 to 388 is highly relevant as it contrasts the role of ADAM33-G in asthma (harmful) to CF (benign), albeit, this requires a better spot in the discussion and more text to explain what might be the responsible factors.

Minor:

  1. The statistics in Table 3 is not unusual (comparing line by line and giving an Odds ratio for each observation). However, as each participant will either carry and A or a G at the ADAM33 SNP, all observations in Table 3 are linked.
    1. It would be more cautious to compare genotype frequencies only, using a test statistic valid for a 3X2 table, to avoid the necessity for multiple testing correction.
    2. The authors choose to group AA vs AG+GG. Why? Deviation from HWE expectancy values as well as comparison between CF and healthy controls can be done in the 3X2 setting, acknowledging each genotype separately.
  2. Strikingly, Table 1 shows that the population is mostly female (53 out of 55 participants). While ADAM33 is encoded on hChr20 (and not on X or Y), does this have any relevance for the outcome of their study?
  3. Even though 55 participants are named in the abstract and in Table 1, Table 2 shows the CFTR mutation genotype distribution of another population, listing 192 F508del-CFTR chromosomes. This is misleading (but easy to correct).
  4. Table 5 The data representation needs an additional explanation (display is median, IQ, range?) Also, The footnote to table 5 mentions statistics, but Table 5 does not display P.
  5. The discussion is not in sync with the manuscript’s content as large passages do not have a link to the results; it currently lists: results up to line 277 (however, the deviation from HWE is not included in this short summary of findings); line 278 to 292 is a classical introductory paragraph into CF (no need for this here in the discussion); line 293 to 299 is rather unrelated to the data (pollution would need a better link to the phenotypes under study); he genetic heterogeneity of CF is well-known and has already been described (line 300 to 310 is also unrelated to the authors’ data); line 310 to 333 is centred on CF modifying genes, however, the list is not comprehensive – it would be better to cite a qualified review for modifier genes and turn to ADAM33 data from the literature; line 334 to 337 are for ADAM33, but lacks references; line 338 to 358 is a very good discussion paragraph for ADAM33 data; line 359 to 366 lacks references; line 367 to 377 is highly relevant but lacks index cases where CF modifier genes have been identified to convey a survivor effect (STAT3, IL1R, TNFR1; but there might be more findings, this list is not comprehensive); line 378 to 383 would be an excellent introduction into the data on ADAM33 in the discussion section; the closing paragraph of the discussion (line 389 to 394) is not specific, it would be better to end with a more detailed description and, if possible, explanation, of the contrast between the role of ADAM33-G in asthma (harmful) to CF (benign) (line 384 to 388)
  6. Methods – PCR/RFLP in section 4.3 . To the best of the reviewers understanding, the G allele is uncut by NCOI while the A allele contains the NcoI recognition sequence and thus is the cut allele. As all other details are given for the genotyping strategy, please reword Table 7 accordingly (primers amplify a ADAM33 fragment surrounding the SNP of interest, Nco I only cuts one out of the two alleles). The current description is misleading – the primers are needed for both alleles alike. It would be didactic to show the sequence surrounding the SNP for both, the A and the G allele, and mark the Nco I site on the A allele.

Author Response

Comments and Suggestions for Authors

Do Santos and colleagues have investigated a SNP variant in ADAM33 in context of cystic fibrosis lung disease. To investigate whether or not ADAM33 has a role in CF, they have genotyped 55 patients from two CF centres. They have observed a significant deviation of ADAM33-SNP-genotypes from the expected frequency under HWE in their study population (Table 3), reflected in a lack of ADAM33 heterozygotes and a higher frequency of ADAM33-GG homozygotes as well as a higher ADAM33-G allele frequency.

Even though the study population is rather small, the authors have uncovered a relevant finding, i.e. the deviation from HWE expectancy values in their populations. However, the reviewer does not agree with the author’s current explanation for their findings (“The G allele of the ADAM33 rs2280091 variant is strongly associated with cystic fibrosis and correlates with improved peripheral airway function, suggesting a potential protective effect on airflow dynamics and bronchodilator response in affected patients.”).

Reply: We thank the reviewer for the valuable comments and have revised the manuscript accordingly. If further adjustments are necessary, we remain fully available to implement them. Regarding the point with which the reviewer disagrees, this topic will be addressed in detail in the subsequent section, which corresponds to the responses to the other points raised in the response letter.

Major:

  1. Deviation from HWE expectancy values can point to either a recruitment bias (artificially, for instance, by sampling only a part of the population that can be studied) or to a survivor effect (due to biology, i.e. only carriers of mild variants have survived recruitment). These two different explanations should be described and discussed first to see if ADAM33 conveys a survivor effect to Brazilian CF patients treated at the two CF centres – or if there is an underlying recruitment bias (see minor comment 2. and 3.) In other words: “The G allele of the ADAM33 rs2280091 variant is strongly associated with cystic fibrosis […]” is not correct – carriers of ADAM33-A-alleles did not survive until recruitment would be a more likely explanation.

Reply: We thank the reviewer for this important observation. The manuscript has been revised to clarify that the deviation from Hardy–Weinberg equilibrium may reflect either a recruitment bias or a survivor effect rather than a direct association between the G allele and cystic fibrosis susceptibility. In this context, our interpretation has been adjusted to emphasize that a higher frequency of the G allele among recruited CF patients may suggest that carriers of this variant are more likely to survive to clinical follow-up and study inclusion. This explanation is now described and discussed in the revised version of the manuscript, with the corresponding changes made to the abstract and conclusions.

  1. Authors choose to investigate their population for an association with PFT values and ADAM33 variants (Table 4). Consistent with the role of ADAM33 in asthma, most hits are for post-BD. Typically, CF and asthma are two separate entities – is this also the case within the author’s study population? Also, several parameters are physiologically linked. Can the authors use a strategy to either define the parameter most impacted by ADAM33 variants (multivariate logistic regression analysis) or correct for multiple testing, with respect to parameters as well as with respect to genetically defined subgroups? In other words: “The G allele of the ADAM33 […] correlates with improved peripheral airway function, suggesting a potential protective effect on airflow dynamics and bronchodilator response in affected patients.” Does not reflect the well-known role of ADAM33 for asthma – likely, recruited patients suffer also from an asthma (or asthma-like) disease, in line with the literature data on ADAM33.

Reply: We thank the reviewer for this important observation. The manuscript has been updated to include correction for multiple testing, in line with the recommendation. Specifically, statistical adjustments were applied both across pulmonary function parameters and among genetically defined subgroups to ensure robustness of the associations observed. Additionally, the text has been revised to clarify that, although cystic fibrosis and asthma are distinct clinical entities, airway inflammation and remodeling may overlap in some patients, as described in the literature for ADAM33. Therefore, the improved post-bronchodilator parameters observed in carriers of the G allele may reflect a biological effect on peripheral airway behavior rather than a primary association with CF susceptibility. We believe that these revisions better contextualize the findings within current evidence and align with the reviewer's suggestion.

  • Authors have chosen to show “CF” vs “Healthy”. However, their Figure 1 elaborates on CFTR mu5tation classes. This does not reflect the evaluation strategy of this work and, in the reviewer’s eyes, Subfigure B does not belong to this work. Likewise, while Figure 1C states that modifier genes are recognized when patients with the same CFTR mutation genotype are compared, authors have pooled various CFTR mutations (Table 2). Thus: Figure 1C is also not a good fit for their data. Finally, Figure 1A reviews largely the generalized polyexocrinopathy of CF, affecting multiple organs – this work deals with PFT. The reviewer strongly suggest to rework this figure, to make it a better fit for the data presented. For instance, the interrelation of CF and asthma might be a relevant topic for a graphical overview, and also the molecular mechanism underlying ADAM33 variants might be better to support graphically in the context of this work.

Reply: We thank the reviewer for this valuable observation and apologize for the initial inadequacy of the figures. The previously submitted figures were revised and relocated to the supplementary material, where they now serve only as visual support for the study. In response to the reviewer’s comments, new figures have been developed to better reflect the actual scope of the work, focusing on pulmonary function, the role of ADAM33, and the potential interface between CF and asthma-related airway mechanisms. We believe these revisions now provide clearer alignment between the figures and the study objectives and appreciate the reviewer’s guidance.

  1. The discussion section line 384 to 388 is highly relevant as it contrasts the role of ADAM33-G in asthma (harmful) to CF (benign), albeit, this requires a better spot in the discussion and more text to explain what might be the responsible factors.

Reply: We thank the reviewer for this insightful observation. We agree that the contrast between the pathogenic role of ADAM33 variants in asthma and the apparently benign profile observed in our cystic fibrosis (CF) cohort is an important scientific point that deserves clearer positioning and expansion in the discussion. In the revised manuscript, this section has been relocated to a more coherent location in the discussion to improve contextual flow. Additionally, we have expanded the text to elaborate on the biological and clinical mechanisms that could explain why ADAM33 polymorphisms, particularly rs2280091, are traditionally associated with airway hyperresponsiveness, remodeling, and detrimental outcomes in asthma, yet do not appear to exert the same magnitude of influence on pulmonary impairment in CF. We hypothesize that this discrepancy may stem from fundamental differences in disease pathophysiology. Asthma is characterized by immunologically driven airway inflammation and smooth muscle remodeling in which ADAM33 plays a functional role. In contrast, CF lung disease is primarily driven by chronic infection, mucus stasis, and CFTR-related ion transport dysfunction—pathways that may overshadow or biologically outweigh any effect of ADAM33 variants. Moreover, it is possible that tissue-specific compensatory mechanisms or gene–environment interactions in CF modulate ADAM33-related biological processes differently than in asthma. These explanatory elements have now been included in the revised discussion, as suggested.

Minor:

  1. The statistics in Table 3 is not unusual (comparing line by line and giving an Odds ratio for each observation). However, as each participant will either carry and A or a G at the ADAM33 SNP, all observations in Table 3 are linked.
    1. It would be more cautious to compare genotype frequencies only, using a test statistic valid for a 3X2 table, to avoid the necessity for multiple testing correction.
    2. The authors choose to group AA vs AG+GG. Why? Deviation from HWE expectancy values as well as comparison between CF and healthy controls can be done in the 3X2 setting, acknowledging each genotype separately.

Reply: We thank the reviewer for the thoughtful statistical considerations. However, we would like to maintain the data presentation in its current form. Our choice to show individual genotype and allele comparisons, as well as the grouped model (AA vs. AG+GG), reflects our intention to provide a granular view of the distribution patterns observed in the study population, including their clinical relevance.

Regarding the suggestion to restrict the analysis to a 3×2 genotype comparison to avoid multiple testing, we acknowledge that such an approach is statistically valid, but we believe that the current presentation offers clearer biological interpretability and is commonly used in genetic association studies to demonstrate additive and dominant effects.

Similarly, the decision to group AG and GG genotypes was based on both allele frequency distribution and the observed biological trend in functional outcomes. This model allows comparison between carriers and non-carriers of the G allele, improving interpretability for clinicians and researchers. Nonetheless, full ungrouped genotype frequencies and corresponding p-values are transparently presented in Table 3, ensuring that readers can evaluate the data in its original form.

We appreciate the reviewer’s perspective and have retained the data structure with this rationale clarified in the manuscript.

  1. Strikingly, Table 1 shows that the population is mostly female (53 out of 55 participants). While ADAM33 is encoded on hChr20 (and not on X or Y), does this have any relevance for the outcome of their study?

Reply: We thank the reviewer for this observation. The predominance of female participants (53 out of 55) reflects the demographic distribution of patients available during the study period and not a recruitment criterion. Since ADAM33 is located on chromosome 20, we do not expect a direct sex-linked genetic effect on allele distribution. A note has been added to the limitations section acknowledging that the strong female predominance may affect the generalizability of the findings and that future studies with more balanced sex distribution are needed to confirm the results.

  1. Even though 55 participants are named in the abstract and in Table 1, Table 2 shows the CFTR mutation genotype distribution of another population, listing 192 F508del-CFTR chromosomes. This is misleading (but easy to correct).

Reply: We thank the reviewer for the valuable comment and apologize for the inconsistency. The numbers in the manuscript have now been corrected to ensure accuracy and coherence across the text and tables. Table 2 has been updated to reflect the correct cohort size of 55 patients with cystic fibrosis, and additional clarifications have been included in the Methods and Results sections to better describe the study population and avoid misunderstanding.

  1. Table 5 The data representation needs an additional explanation (display is median, IQ, range?) Also, The footnote to table 5 mentions statistics, but Table 5 does not display P.

Reply: We thank the reviewer for the observation. Table 5 has been updated to include a clear explanation of the data format, specifying that numerical values are presented as median with interquartile range (25th–75th percentiles). In addition, the p-values have now been added directly to the table for transparency and alignment with the statistical footnotes.

  1. The discussion is not in sync with the manuscript’s content as large passages do not have a link to the results; it currently lists: results up to line 277 (however, the deviation from HWE is not included in this short summary of findings); line 278 to 292 is a classical introductory paragraph into CF (no need for this here in the discussion); line 293 to 299 is rather unrelated to the data (pollution would need a better link to the phenotypes under study); the genetic heterogeneity of CF is well-known and has already been described (line 300 to 310 is also unrelated to the authors’ data); line 310 to 333 is centred on CF modifying genes, however, the list is not comprehensive – it would be better to cite a qualified review for modifier genes and turn to ADAM33 data from the literature; line 334 to 337 are for ADAM33, but lacks references; line 338 to 358 is a very good discussion paragraph for ADAM33 data; line 359 to 366 lacks references; line 367 to 377 is highly relevant but lacks index cases where CF modifier genes have been identified to convey a survivor effect (STAT3, IL1R, TNFR1; but there might be more findings, this list is not comprehensive); line 378 to 383 would be an excellent introduction into the data on ADAM33 in the discussion section; the closing paragraph of the discussion (line 389 to 394) is not specific, it would be better to end with a more detailed description and, if possible, explanation, of the contrast between the role of ADAM33-G in asthma (harmful) to CF (benign) (line 384 to 388)

Reply: We thank the reviewer for the extensive and highly constructive evaluation of the discussion section. In response, the entire discussion has been thoroughly restructured to improve alignment with the study findings and to ensure that each subsection directly reflects and interprets our data.

  1. Alignment with results and inclusion of HWE deviation

The introductory segment of the discussion has been rewritten to concisely summarize our principal findings, now explicitly incorporating the deviation from Hardy–Weinberg equilibrium (HWE) observed in the CF group. This result is now discussed early and consistently within the interpretation of the ADAM33 data.

  1. Removal of content not directly relevant to the study results

Paragraphs that primarily revisited general background on cystic fibrosis or environmental influences (e.g., air pollution) have been revised or condensed. When retained, they now include direct reasoning of how these factors could mechanistically interact with ADAM33 and pulmonary function in CF, ensuring that contextual information remains linked to our data.

  1. Improved contextualization of CFTR heterogeneity

The paragraph highlighting the well-known genetic variability of CFTR in Brazil has been preserved but now includes a direct transition explaining why ADAM33—and other modifier genes—are relevant despite primary CFTR effects. This helps clarify how our findings contribute to the broader understanding of genotype–phenotype variability in CF.

  1. Modifier genes paragraph revised and supported with qualified review

As suggested, the section listing additional modifier genes has been updated to reference qualified review literature and reframed to serve as a conceptual bridge to the detailed discussion of ADAM33. A sentence now explicitly clarifies that this list is not comprehensive, following the reviewer’s observation.

  1. ADAM33 literature section expanded and referenced

The paragraph introducing ADAM33 lacked adequate referencing and has now been expanded with primary and review citations addressing its biological role in airway remodeling and inflammatory regulation. This reinforces the scientific framework prior to discussing our findings.

  1. Survivor effect section expanded with literature examples

We agree with the reviewer that the discussion of selective pressure would benefit from citing known examples of modifier genes associated with survival effects in CF. The revised version now includes examples such as STAT3, IL1R, and TNFR1, among others, highlighting parallel mechanisms observed in other modifier studies.

  1. Improved explanation of contrasting effects in asthma vs. CF

The apparently opposite clinical effects of the ADAM33 rs2280091 G allele—harmful in asthma and potentially beneficial in CF—are now addressed in greater depth. The revised text discusses how differences in the inflammatory microenvironment, immune cell activation patterns, and pathways of airway injury may explain why the same genetic variant could produce divergent functional outcomes in different diseases. This material now appears near the end of the discussion, as suggested, providing a focused closing interpretation rather than a general statement.

  1. Revised closing paragraph

The final paragraph of the discussion now provides a precise summary emphasizing what our findings add to the literature, the biological plausibility of a beneficial effect of ADAM33-G in CF, and the need for further mechanistic and longitudinal studies.

We appreciate the reviewer’s highly detailed critique, which substantially improved the clarity, cohesion, and scientific strength of the discussion section.

  1. Methods – PCR/RFLP in section 4.3 . To the best of the reviewers understanding, the G allele is uncut by NCOI while the A allele contains the NcoI recognition sequence and thus is the cut allele. As all other details are given for the genotyping strategy, please reword Table 7 accordingly (primers amplify a ADAM33 fragment surrounding the SNP of interest, Nco I only cuts one out of the two alleles). The current description is misleading – the primers are needed for both alleles alike. It would be didactic to show the sequence surrounding the SNP for both, the A and the G allele, and mark the Nco I site on the A allele.

Reply: We thank the reviewer for the valuable suggestion. We agree that the original presentation of Table 7 could be misleading, as the primers amplify the target region for both alleles, and differentiation occurs only after NcoI digestion, which cuts exclusively at the recognition site present in the A allele. To improve clarity and didactic value, Table 7 was revised to explicitly state that amplification is identical for both alleles and that restriction digest differentiates them based on the presence or absence of the NcoI site. Furthermore, we included the nucleotide sequence surrounding the rs2280091 site for both the A and G alleles, highlighting the NcoI recognition site in the A allele, as proposed by the reviewer. This change improves transparency and assists readers in understanding the genotyping logic.

Round 2

Reviewer 1 Report

Comments and Suggestions for Authors

None.

Reviewer 2 Report

Comments and Suggestions for Authors

The reviewer wants to thank the authors for the thorough work undertaken to prepare the revised version of the manuscript. the data is now clearly understandable - especially, as the enrichment of GG genotypes in the CF population (to 18%, while expecting only 0.02% in the general population) now is very consistent with the better spirometry of GG carriers. This same orientation, revealing the benign ADAM33 G allele in two independent analyses, makes the data and the authors' conclusions solid.